# Small Hero with Great Powers: Vaccinia Virus E3 Protein and Evasion of the Type I IFN Response

**DOI:** 10.3390/biomedicines10020235

**Published:** 2022-01-22

**Authors:** Mateusz Szczerba, Sambhavi Subramanian, Kelly Trainor, Megan McCaughan, Karen V. Kibler, Bertram L. Jacobs

**Affiliations:** 1Biodesign Center for Immunotherapy, Vaccines and Virotherapy, Arizona State University, Tempe, AZ 85281, USA; mateusz.szczerba@asu.edu (M.S.); sambhavi.subramanian@gmail.com (S.S.); kelly.trainor@coconino.edu (K.T.); megan.mccaughan@asu.edu (M.M.); karen.kibler@asu.edu (K.V.K.); 2School of Life Sciences, Arizona State University, Tempe, AZ 85281, USA; 3Vir Biotechnology, San Francisco, CA 94158, USA; 4Faculty of Biology, Coconino Community College, Flagstaff, AZ 86005, USA

**Keywords:** interferon, vaccinia virus, host range, PKR, necroptosis, Z-RNA, ZBP1, pathogenesis, innate immunity, viral immune evasion

## Abstract

Poxviridae have developed a plethora of strategies to evade innate and adaptive immunity. In this review, we focused on the vaccinia virus E3 protein, encoded by the *E3L* gene. E3 is present within the *Chordopoxvirinae* subfamily (with the exception of the avipoxviruses and molluscum contagiosum virus) and displays pleiotropic effects on the innate immune system. Initial studies identified E3 as a double-stranded RNA (dsRNA)-binding protein (through its C terminus), able to inhibit the activation of protein kinase dependent on RNA (PKR) and the 2′5′-oligoadenylate synthetase (OAS)/RNase L pathway, rendering E3 a protein counteracting the type I interferon (IFN) system. In recent years, N-terminal mutants of E3 unable to bind to Z-form nucleic acids have been shown to induce the cellular death pathway necroptosis. This pathway was dependent on host IFN-inducible Z-DNA-binding protein 1 (ZBP1); full-length E3 is able to inhibit ZBP1-mediated necroptosis. Binding to what was identified as Z-RNA has emerged as a novel mechanism of counteracting the type I IFN system and has broadened our understanding of innate immunity against viral infections. This article gives an overview of the studies leading to our understanding of the vaccinia virus E3 protein function and its involvement in viral pathogenesis. Furthermore, a short summary of other viral systems is provided.

## 1. Introduction

The family of *Poxviridae* encompasses a group of large (200 nm × 300 nm), brick-shaped, enveloped viruses that are known to infect both invertebrates (subfamily *Entomopoxvirinae*) and vertebrates (subfamily *Chordopoxvirinae*) [1,2,3]. Poxviruses replicate entirely in the cytoplasm, which distinguishes them from most other DNA viruses. The poxviral genome is a single, linear, double-stranded DNA (dsDNA) molecule, which ranges in size from 130 to 300 kbp [4]. In its conserved central region, the poxviral genome encodes proteins essential for virus replication, such as transcription and DNA replication enzymes. The more variable terminal regions are known to encode non-essential virulence factors, proteins affecting host range, and proteins involved in modulating the host’s innate immune response [4,5,6].

The most studied member of the *Poxviridae* family is vaccinia virus (VACV), which is the live vaccine used to immunize against and eradicate smallpox [7]. Vaccinia virus belongs to the orthopoxvirus genus, which encompasses other notable members: for instance, variola virus (the causative agent of human smallpox), monkeypox virus, and cowpox virus (known human and animal pathogens) [5,7,8].

Poxviruses are known to be resistant to the activity of type I interferons (IFNs) [9]. IFNs are the hallmark effector of the host’s response during viral infection. IFN’s function is to initiate the innate immune response, mediate the development of the adaptive immune response, and upregulate the expression of IFN-stimulated genes (ISGs) to establish an antiviral state [10]. Viral pathogen-associated molecular patterns (PAMPs) are sensed in the cell by pattern recognition receptors (PRRs): Toll-like receptors (TLRs), retinoic acid-inducible gene I (RIG-I)-like receptors (RLRs), and cytosolic sensors, including Z-DNA-binding protein 1 (ZBP1) and cyclic GMP-AMP synthase (cGAS) (Figure 1) [11,12,13,14,15,16,17]. Sensing of viral PAMPs by PRRs induces a signaling cascade leading to activation of transcription factors, such as IFN-regulatory factor 3 (IRF3) and nuclear factor kappa-light-chain-enhancer of activated B cells (NFκB), which localize into the nucleus and induce expression of type I IFN genes [18,19,20,21]. Produced IFNs will be excreted from the cell and bind to type I IFN receptors (IFNAR) on neighboring cells, activating the Janus kinase (JAK)/signal transducer and activator of transcription (STAT) signaling pathway [20,22,23]. Within this cascade, IFN-stimulated gene factor 3 (ISGF3) complex, comprising STAT1, STAT2, and IRF9, upregulates expression of ISGs, including protein kinase dependent on RNA (PKR) and 2′5′-oligoadenylate synthetase (2′5′-OAS). Expressed PKR becomes activated by dsRNA produced during poxviral infection, which results in phosphorylation of the eukaryotic translation initiation factor eIF2 at its α subunit, and inhibition of protein synthesis (Figure 1) [24,25,26,27]. Furthermore, dsRNA can also activate 2′5′-OAS, which has the ability to produce unusual 2′-5′-oligoadenylates, leading to activation of RNase L, which in turn degrades both host and viral mRNAs [28,29,30].

The cellular ZBP1 protein is a known cytosolic sensor of DNA and its expression is induced as one of the ISGs during viral infection [31]. Upon sensing of DNA in the cytoplasm, ZBP1 induces a signaling cascade leading to the production of type I IFN [31,32]. Aside from its B-DNA-binding domain allowing ZBP1 to bind to the canonical form of DNA, ZBP1 also contains two Z-DNA-binding domains (ZBDs): Zα1 and Zα2. Therefore, ZBP1 is also known as a Z-DNA-binding protein and has been shown to bind to the alternative, left-handed, less compact form of DNA, Z-DNA [32,33,34]. Z-DNA has been identified using specific antibodies in areas of the genome that undergo active transcription [35,36]. It is believed that this less compact form of DNA facilitates recruitment of DNA-dependent RNA polymerase. Furthermore, Z-DNA-binding proteins, ZBP1 and RNA-specific adenosine deaminase 1 (ADAR1) are found in cellular structures referred to as stress granules [37,38,39]. Nonetheless, the function of Z-DNA-binding proteins in these granules has not yet been determined.

In recent years, a new function has been ascribed to ZBP1 as a mediator of a programmed cell death (PCD) pathway called necroptosis ([16,40,41,42,43,44], described in this review). PCD takes place in an organized manner, following a signaling cascade, which is specifically induced and targets a specific outcome. Apoptosis is executed by the Bcl-2 protein family (only the intrinsic pathway, induced by DNA damage) and cysteinyl aspartate-specific proteases (caspases) (both the intrinsic and the extrinsic pathway, induced following the stimulation of a death receptor, e.g., tumor necrosis factor receptor, TNFR) [45,46,47,48,49]. Apoptosis is immunologically silent as it leads to membrane blebbing and the production of apoptotic bodies, which are subsequently engulfed by surrounding cells and phagocytes [50]. On the contrary, pyroptosis and necroptosis are highly inflammatory as they lead to the release of the cellular contents directly into the extracellular matrix, which induces inflammation [51,52]. Pyroptosis is induced by the stimulation of NOD-like receptors (NLRs), which leads to the activation of caspase-1 and the formation of the inflammasome [53]. Canonically, necroptosis is induced following the stimulation of a death receptor (like TNFR), which activates receptor-interacting protein kinase 1 (RIPK1), which can in turn associate with RIPK3 via mutual RIP homotypic interaction motifs (RHIMs), thus, leading to RIPK3 phosphorylation and subsequent phosphorylation of mixed lineage kinase-like (MLKL) protein [54,55,56,57]. Gasdermin D, which is activated by caspase-1, and MLKL, activated by RIPK3, localize to the plasma membrane where they form pores, which results in dissipation of the cellular ionic gradient, water influx, cell swelling, and osmotic lysis [58,59,60]. These PCD pathways play a major role in innate immunity.

There are several mechanisms by which poxviruses are able to inhibit the type I IFN response. They are reviewed in [61]. This review focuses on vaccinia virus E3 protein and a novel way of affecting immune response by inhibition of necroptosis, which is dependent on the IFN-inducible ZBP1 protein. We provide an historical overview of how E3 protein was discovered and outline the progress during which E3 function was discerned.

## 2. Early Studies on Poxviruses and the Interferon System

### 2.1. Discovery of E3 Protein

Very early on in the study of the interferon system, it became clear that viruses differed in their sensitivity to interferon. Vaccinia virus (VACV) was shown by the Youngner group to be quite resistant to the pre-treatment of cells with type I IFN [62]. Not only was VACV IFN^R^, but it rescued both vesicular stomatitis virus (VSV) and encephalomyocarditis virus (EMCV) from the effects of IFN [9,62,63]. For rescue of EMCV, rescue correlated with inhibition of PKR activation by VACV [9].

Whitaker-Dowling first demonstrated that VACV encoded an inhibitor of the IFN-inducible antiviral protein kinase, PKR [64]. Our group purified this PKR-inhibitor from infected cells, obtained protein sequence information for the putative inhibitor, and identified it as the product of the VACV *E3L* gene [65]. Isolated E3 protein had potent PKR-inhibitory activity, and VACV deleted of *E3L* (vp1080) failed to inhibit PKR activation (Figure 1 and Figure 2) [65,66]. Unlike wt VACV, VACV vp1080 failed to replicate in most cell lines in culture and was IFN^S^ in rabbit RK-13 cells where it did replicate [67,68]. VACV vp1080 failed to rescue VSV from the antiviral effects of IFN, but still efficiently rescued EMCV, suggesting the presence of a second IFN^R^ gene [69]. This was identified as K3L, which encodes an eIF2α homologue, which could inhibit PKR or any of the other eIF2α protein kinases [70,71]. K3L partially rescued EMCV from the effects of IFN but had no effect on IFN^S^ of VSV [69]. Likewise, VACV deleted of K3L (vp872) failed to rescue EMCV, but efficiently rescued VSV from the effects of IFN [69]. Thus, *E3L* was essential for the rescue of VSV, but dispensable for the rescue of EMCV, while K3L led to at least partial rescue of EMCV, but had no effect on the rescue of VSV from the effects of IFN, suggesting a complementary function of the two proteins against the host’s IFN system.

### 2.2. C Terminus of E3 Contains a dsRNA-Binding Domain

When *E3L* was first identified as the primary IFN^R^ gene of vaccinia virus, a BLAST search identified the C-terminal half of the protein as having sequence similarity with proteins known to bind to dsRNA, such as PKR and RNase III [66]. We were able to show that *E3L*-encoded proteins bound specifically to dsRNA, but not single-stranded RNA (ssRNA) or dsDNA, and that this conserved domain was necessary and sufficient for binding (Figure 2 and Table 1) [72]. IFN^R^ and the broad host range of VACV also mapped to this dsRNA binding motif. Furthermore, a virus missing the entire N-terminal half of *E3L* (VACV *E3L*Δ83N) was IFN^R^ in RK-13 cells and replicated in a wide variety of human and monkey cells [73,74] despite leading to the activation of PKR and OAS, albeit at very late times post-infection. Surprisingly, both the C-terminal dsRNA-binding domain, which is essential for E3 function and the N-terminal domain, which is dispensable for IFN^R^ and a broad host range, are conserved amongst the majority of poxviruses that infect mammals [75]. Furthermore, the distantly related ORF virus *E3L* homologue could fully replace VACV *E3L* for IFN^R^ and replication in most cells in culture [74]. This shows that while we could not detect an essential function for the N-terminal domain in rabbit, human, or monkey cells in culture, it was in fact conserved amongst poxviruses, suggesting an important role in vivo. While the N-terminal domain of E3 was conserved amongst most mammalian poxviruses, it showed no homology with other proteins in available databases.

## 3. E3 Protein Function in Viral Pathogenicity: Studies In Vivo

### 3.1. E3 Protein Is VACV Virulence Factor

Since the N-terminal domain of E3 was dispensable for IFN^R^ in RK-13 cells and for replication in a wide variety of human and monkey cells, despite being evolutionarily conserved, we decided to ask if this domain was necessary for pathogenesis in the mouse model. VACV WR is a mouse-adapted VACV strain that has an intranasal (IN) LD_50_ of 10^3–^10^4^ pfu, depending on the strain of mouse [76,77]. VACV WR Δ*E3L* is highly attenuated in the mouse model, having an LD_50_ greater than 10^7^ pfu. Both the C-terminal dsRNA-binding domain and the conserved N-terminal domain are necessary for pathogenesis since both VACV WR *E3L*Δ83N and VACV WR *E3L*Δ26C (which has no detectable affinity for dsRNA) are highly attenuated (Table 1) [76]. This was the first suggestion that the conserved N-terminus of *E3L* was biologically relevant.

### 3.2. E3 Protein Is a Type I IFN Antagonist

To begin to understand the function of E3 in vivo, we infected IFNAR^−/−^ mice with VACV WR Δ*E3L* and VACV WR *E3L*Δ83N. While the lack of an IFN signaling system did not restore pathogenesis to VACV WR Δ*E3L* when animals were infected IN, it partially restored pathogenesis upon intracranial (IC) infection [77]. This is likely due to the lower dose of virus needed to induce pathogenesis when the virus is administered IC. Only partial restoration of pathogenesis in IFNAR^−/−^ mice suggests that there is a non-IFN-inducible pathway that can at least partially inhibit pathogenesis induced by VACV WR Δ*E3L*. VACV WR Δ*E3L* also gains virulence in MDA5^−/−^, MAVS^−/−^, and IRF3^−/−^ mice, consistent with a key role for the IFN system in blocking pathogenesis by VACV WR Δ*E3L* [78]. Pathogenesis of VACV WR *E3L*Δ83N was fully restored in IFNAR^−/−^ mice when the virus was administered either IN or IC [77]. This suggested that the sole function of the conserved N-terminal domain in vivo is to counteract the type I IFN system.

## 4. E3 Protein Binds to Z-Form Nucleic Acids

### 4.1. Discovery of Z-DNA

In the late 1990s, Alex Rich’s lab screened a chicken cDNA library for proteins that could bind to Z-DNA [79]. They cloned out the IFN-inducible form of the host protein ADAR1. ADAR1 is an RNA modifying enzyme that can deaminate adenosine in dsRNA or RNA with a secondary structure [80,81]. The Rich lab mapped the domain that could bind Z-DNA (Zα domain) to an N-terminal region found only in the IFN-inducible p150 form of ADAR1 (Figure 3) [82,83]. The constitutively expressed form of ADAR1, p110, is lacking this domain. Both p110 and p150 contain a second Zα-related domain (Zβ) that, however, has a key amino acid change that prevents interaction with Z-DNA [75,84].

### 4.2. The N Terminus of E3 Is a ZBD and Binding to a Z-NA Is Required for VACV Pathogenesis

A database search at the time showed a similarity between the Zα domain of ADAR1 to one other IFN-inducible protein, ZBP1—which at the time was known as DLM-1 and later renamed DAI (ZBP1 contains two Zα domains, Zα1 and Zα2)—and to the conserved N-terminal domain of poxvirus E3 protein, that we showed is essential for pathogenesis in mice (Figure 3) [34,76,85,86]. This allowed us to collaborate with the Rich lab to ask if the capacity to bind to Z-DNA is critical for VACV to be pathogenic in mice. We first made chimeric viruses, replacing the N-terminal domain of E3 protein with either the Zα or Zβ domain of ADAR1, or the Zα1 domain of ZBP1 (Figure 2). Replacing the N-terminus of E3 with either of the two Z-DNA binding domains—the Zα domain of ADAR1 or the Zα1 domain of ZBP1—led to viruses with virulence indistinguishable from wt VACV WR, while replacing the N-terminus of E3 with the related Zβ domain of ADAR1, which cannot bind Z-DNA, led to a virus that was attenuated to a similar degree as *E3L*Δ83N virus, which completely lacks the conserved N-terminus (Table 1) [75]. This allowed us to perform a structure/function analysis based on the crystal structure of the Zα domain of ADAR1 bound to Z-DNA [87]. Mutation of residues that contacted Z-DNA in crystal structures led to proteins with reduced affinity for Z-DNA and led to a virus that was attenuated in vivo [75]. Furthermore, mutation of several residues not forming contacts with Z-DNA in the crystal structure had wt, nanomolar affinity for Z-DNA, and wt virulence in vivo. Equivalent mutations in authentic E3 protein led to viruses with equivalent virulence in vivo [75]. As mentioned above, the Zβ domain of ADAR1 cannot bind Z-DNA [84]. In the Zβ domain of ADAR1, residue 335, which in the equivalent position in Zα domains is a conserved tyrosine (Y177 in the Zα domain of ADAR1), has been changed to an isoleucine. Residue Y177 makes several key interactions with Z-DNA in the crystal structure and, when mutated, attenuates the virus. A mutation of isoleucine 335 to tyrosine (I335Y) partially restored binding to Z-DNA, and partially restored virulence in the virus expressing a chimeric ZβI335Y E3 protein [75]. Thus, all of our data is consistent with the binding of E3 protein to a Z-form nucleic acid in infected cells being critical for VACV pathogenesis in mice.

## 5. Zα Domain of E3 Plays a Role in Inhibition of Necroptosis

### 5.1. N Terminus of E3 Is Required for Full PKR Inhibition in 129 MEFs

The identification of both a biological function in vivo for the N-terminus (IFN^R^) and a biochemical function (binding to Z-DNA) led us to redouble efforts to find cells in which VACV *E3L*Δ83N had a phenotype. We discovered three murine cells in culture in which VACV *E3L*Δ83N had a phenotype: 129 MEFs, L929 cells, and JC cells. IFN sensitivity of VACV *E3L*Δ83N in 129 MEFs did not correlate with the binding of E3 proteins to Z-DNA but instead correlated with the inability to fully sequester dsRNA [77]. In addition to not binding to Z-DNA, VACV *E3L*Δ83N cannot fully inhibit PKR activation in cells in culture [26]. The residues necessary to fully inhibit PKR are different from the residues necessary for Z-DNA binding. A failure to fully inhibit PKR by N-terminal mutants of E3 is due to a failure to fully sequester dsRNA in infected cells [26,72,74]. In 129 MEFs, IFN^R^ of VACV *E3L*Δ83N could be fully recovered in PKR^−/−^ cells or by PKR knock-down [77]. Additionally, selection of a virus that makes much less dsRNA than wt VACV fully rescued IFN^R^ of an N-terminal mutant of E3 [88]. However, the pathogenesis of VACV *E3L*Δ83N was not restored in PKR^−/−^ mice [77]. This suggested that an additional IFN-inducible function, perhaps in addition to activation of PKR, is important for the attenuation of VACV *E3L*Δ83N in vivo.

### 5.2. Binding to Z-NAs by E3 Is Essential for VACV IFN Resistance

In mouse L929 cells, VACV *E3L*Δ83N is IFN^S^ while wt VACV is IFN^R^, and in mouse JC cells VACV *E3L*Δ83N fails to replicate, while wt VACV replicates efficiently (see Figure 4) [40,88]. Unlike replication in 129 MEFs, IFN^R^ in L929 cells and replication in JC cells correlate with Z-DNA binding among mutants of *E3L* (Figure 4) [16,88]. Thus, the phenotype in N-terminal mutants of E3 in these cells correlates with the phenotype seen in vivo (see Section 4.2 and Table 1).

### 5.3. Necroptosis Inhibition Is Conferred by N Terminus of E3

In mouse L929 cells, the infection of IFN-treated cells with VACV *E3L*Δ83N leads to rapid explosive cell death, by 6 h post-infection. Treatment with a pancaspase inhibitor, Z-VAD-FMK (zVAD), did not rescue either cell death or IFN^R^ of VACV *E3L*Δ83N [40]. This suggested that cell death is caspase-independent and thus, likely not apoptosis or pyroptosis [89]. Necroptosis, which is a caspase-independent form of programmed cell death, requires the cellular kinase RIPK3 to phosphorylate MLKL, the executioner of necroptotic cell death, on Ser358 (Ser345 in mice) [55,90,91,92,93]. RIPK3 kinase activity can be inhibited by the small molecule GSK872. GSK872 treatment inhibits VACV *E3L*Δ83N-induced cell death and fully rescues VACV *E3L*Δ83N from the effects of IFN-treatment [40]. VACV *E3L*Δ83N infection of IFN-treated L929 cells leads to phosphorylation of MLKL that is inhibitable by GSK872 [40]. Phosphorylated MLKL can migrate to the plasma membrane, trimerize and form pores in the membrane that allow fluid influx, swelling and bursting [59,60,94,95]. MLKL trimerizes in VACV *E3L*Δ83N-infected cells and trimerization is blocked by GSK872 treatment [40]. CRISPR knockout of MLKL fully rescues VACV *E3L*Δ83N from the effects of IFN [96]. Thus, VACV *E3L*Δ83N induces necroptotic cell death in IFN-treated L929 cells that is associated with the inhibition of virus replication.

### 5.4. ZBP1 Mediates Necroptos Is Triggered by a VACV N-Terminal Mutant of E3

Any of the three sensors can lead to the activation of RIPK3. RIPK3 contains an RHIM that allows it to interact with other RHIM-containing proteins, including Toll/IL-1 receptor domain-containing adapter inducing IFN-β (TRIF), RIPK1, and ZBP1 (also known as DLM-1 and DNA-dependent activator of IFN-regulatory factors (DAI)) (Figure 3) [44,57,97,98,99]. Furthermore, TRIF acts downstream of TLR3 and TLR4, RIPK1 acts downstream of death receptors, while ZBP1 senses Z-form nucleic acid in virus-infected cells [16,43,97,100]. Since loss of the Zα—Z-nucleic acid binding domain—of E3 leads to activation of RIPK3, and since ZBP1 is highly IFN-inducible in L929 cells, we hypothesized that ZBP1 was likely a sensor detecting VACV *E3L*Δ83N infection [40]. CRISPR knockout of ZBP1 prevented VACV *E3L*Δ83N-induced cell death and rescued VACV *E3L*Δ83N from the effects of IFN [96]. In cells that do not express ZBP1, ectopic expression of ZBP1 led to an inhibition of replication of VACV *E3L*Δ83N [16,40].

### 5.5. The ZBD of E3 Interacts with Z-RNA to Inhibit Necroptosis

Data demonstrating sensing of VACV *E3L*Δ83N infection by ZBP1 suggests that the Zα domains of E3 and ZBP1 may be competing with each other for binding to a Z-form nucleic acid in infected cells. Furthermore, Z-form NA can be detected by immunofluorescence (IF) in VACV-infected but not uninfected cells [16]. By IF, Z-NA levels were highest in VACV *E3L*Δ*E3L*-infected cells, followed by VACV *E3L*Δ83N with the lowest amounts in wt VACV-infected cells. Z-NA staining was abolished by pre-treatment with RNase, but not DNase, suggesting that Z-RNA is present in VACV-infected cells [16]. This suggests that a functional N-terminus may be sequestering Z-RNA. This was confirmed by flow cytometry, where Z-RNA was detectable in VACV *E3L*Δ83N, but not wt VACV-infected cells. Over-expression of ZBP1 masked Z-RNA detection in VACV *E3L*Δ83N-infected cells, which was consistent with ZBP1 and the Zα domain of E3 competing for binding to the same Z-RNA [16]. This is also consistent with our data in vivo with chimeric viruses, where the Zα domain of ADAR1 or the Zα1 domain of ZBP1 could functionally replace the Zα domain of E3 protein [75]. We have recently shown that the Zα domain of ADAR1 and the Zα1 or Zα2 domain of ZBP1 (Figure 3) can functionally replace the Zα domain of E3 protein in L929 cells in culture, preventing sensing of VACV infection by ZBP1 [16].

## 6. Role of the ZBP1/RIPK3/MLKL Axis in Pathogenesis In Vivo

ZBP1-Mediated Necroptosis Underlies Attenuation of VACV E3LΔ83N In Vivo

To ask if the role that we have seen in necroptosis in cells in culture correlates with the role of the N-terminus in pathogenesis in mice, we have evaluated the pathogenesis of VACV *E3L*Δ83N in ZBP1^−/−^ and RIPK3^−/−^ mice. In wt mice infected IN, VACV *E3L*Δ83N replicates to a high titer in the nose, but spreads poorly or replicates poorly at distal sites, including the lung, brain, and spleen. The spread is rescued to a large extent in IFNAR^−/−^ mice [77]. In wt mice, VACV *E3L*Δ83N is attenuated by at least four logs on IN infection, and virulence, too, is rescued in IFNAR^−/−^ mice [77]. Consistently, VACV *E3L*Δ83N is further attenuated by IFN treatment of wt mice prior to infection (Figure 5A). To a large extent, attenuation, the spread to distal organs, and IFN resistance is rescued in both ZBP1^−/−^ and RIPK3^−/−^ mice, indicating that much of the IFN sensitivity of VACV *E3L*Δ83N in vivo is likely due to induction of the ZBP1/RIPK3 axis resulting in necroptosis (Figure 5A,B) [40]. Furthermore, VACV *E3L* P63A, a point mutant unable to bind to Z-NA and known not to inhibit necroptosis in cells in culture (Table 1), displays decreased pathogenicity in wt mice, but its pathogenicity is rescued in ZBP1^−/−^ and RIPK3^−/−^ mice (Figure 5C). Furthermore, we demonstrated that infection of wt mice IN with VACV *E3L*Δ83N resulted in the induction of necroptosis, which was not observed in mice infected with wt VACV or uninfected mice (Figure 5D) [101].

## 7. ZBP1 Binding of Z-Form NAs Plays a Role in Pathogenesis in Other Viruses

### 7.1. Cytomegaloviruses

One of the first insights into necroptosis being involved in antiviral immunity comes from studies done by Ed Mocarksi’s group showing that murine cytomegalovirus (MCMV)-encoded M45 protein had the ability to block RIPK3-mediated necroptosis [102]. Infection of cells with MCMV-M45mutRHIM mutant led to induction of necroptosis, which was independent of RIPK1 as necrostatin-1 (RIPK1 inhibitor) and RIPK1-specific shRNAs had no effect on cell death induced by the mutant virus [103]. MCMV-M45mutRHIM induced necroptosis only in cells expressing ZBP1, which was the first time ZBP1 was implicated in this cell death pathway [44]. Furthermore, MCMV-M45mutRHIM is attenuated in wt mice, but not in RIPK3^−/−^ or ZBP1^−/−^ mice [44,103]. These studies demonstrate that modulation of necroptosis is crucial for successful viral pathogenesis, which remains consistent with what we see for VACV *E3L*Δ83N (Table 1) [40].

Maelfait and colleagues established the importance of binding to Z-NAs for necroptosis induced during MCMV infection. MCMV-M45mutRHIM was only able to induce necroptosis in the presence of wt ZBP1, but not when Zα1 and Zα2 domains of ZBP1 were mutated [42]. Furthermore, MCMV-M45mutRHIM infection is successfully cleared in wt mice, but not in knock-in mice expressing ZBP1 with mutated Zα domains, suggesting a role for Z-NAs in clearing MCMV infection. Moreso, ZBP1 was found associated with RNA during MCMV-M45mutRHIM infection and induction of cell death required RNA synthesis but not viral DNA replication, implicating a role for Z-RNA in virally induced necroptosis, similarly to what we suggest for VACV *E3L*Δ83N [16,42]. For both viruses, these Z-RNAs have been shown to be of viral origin [16,104]. Interestingly, human cytomegalovirus (HCMV) displays species-dependent regulation of the necroptosis pathway. HCMV protein UL36 binds to MLKL, leading to MLKL degradation and inhibition of necroptosis in human cells, but not in murine cells (Muscolino et al. 2021).

### 7.2. Herpes Simplex Virus 1

Aside from cytomegaloviruses, other herpesviruses are known to modulate necroptosis, and this modulation is species-dependent. Human herpes simplex virus 1 (HSV-1) infection is successfully cleared in wt mice but not in RIPK3^−/−^ mice [56]. In murine cells, HSV-1 induces necroptosis, which is independent of ZBP1 but depends on viral ribonucleotide reductase large subunit (IC6), which interacts in an RHIM-mediated manner with RIPK1 and RIPK3. IC6 facilitates and is required for RIPK1-RIPK3 and RIPK3-RIPK3 interaction and induction of necroptosis in mice [56,105]. In contrast, IC6 has been shown to inhibit necroptosis in human cells by impairing RIPK1-RIPK3 necrosome formation and reduction of RIPK1 autophosphorylation, suggesting that HSV-1 evolved to escape necroptosis in humans for the sake of establishing a successful infection [106,107,108]. Interestingly, when the RHIM of IC6 is mutated, HSV-1 is still able to induce necroptosis in mice, which this time is dependent on ZBP1. IC6mutRHIM virus infection is also cleared in wt mice, but not in ZBP1^−/−^ mice [109]. Strikingly, ZBP1-mediated necroptosis is also induced in human cells infected with IC6mutRHIM virus, showing that the ZBP1-mediated pathway is species-independent [109].

### 7.3. Influenza A Virus

Z-NAs have been shown to play a role in the pathogenesis of RNA viruses as well. Influenza A virus (IVA) infection is sensed by ZBP1, which activates RIPK3 and triggers MLKL-mediated necroptosis and FADD-mediated apoptosis [43,110]. ZBP1 senses viral ribonucleoprotein (vRNP) complexes, including IAV genomic RNA, nucleoprotein (NP), and polymerase subunit (PB1), and cells lacking ZBP1 or expressing Zα mutants of ZBP1 fail to trigger IAV-induced necroptosis or apoptosis, indicating that viral genomic Z-RNA plays a role [41,43,111]. ZBP1-deficient mice succumb to uncontrolled IAV infection [43]. In other studies, ZBP1^−/−^ and MLKL^−/−^ mice displayed reduced inflammatory response and epithelial damage, demonstrating the pro-inflammatory nature of the necroptotic cell death [41,112]. Interestingly, ZBP1 sensing of viral Z-RNA takes place in the nucleus and it requires ZBP1 ubiquitination induced by IAV infection [111,112].

## 8. Conclusions and Outlook

Vaccinia virus E3 protein is an immune evasion protein conserved among poxviruses [72,75]. E3 displays a pleiotropic effect, with an ability to counteract multiple arms of the host innate immunity: PKR, OAS/RNase L, STING, RIG-I/MDA5, RNA Pol III, TLR2/-8/-9 (described in [61]). With the development of our understanding of PCD pathways, a new function for E3 protein has been revealed as an inhibitor of necroptotic cell death. Additionally, the function of E3 ZBD has been characterized in more detail and added to the progress in understanding the biological function of Z-NAs and the relevant associated proteins, such as ZBP1. Studies described here shed light on the biological significance of Z-NAs, demonstrating their involvement in viral pathogenesis and host innate immunity. The same studies also showed how viruses have evolved to counteract these Z-NA-mediated responses in order to evade host immunity.

VACV E3 protein is the first case of a poxviral protein that has been shown to bind to Z-RNA and inhibit necroptosis (Figure 1). More studies are needed to show if other proteins with similar abilities are present in the poxviral family and other viral families. There are still a few unknown aspects of the necroptosis pathway triggered by Z-RNA that need further investigation. It remains unknown what kind of RNA is responsible for necroptosis induction during VACV *E3L*Δ83N infection. It is unclear whether it is an RNA of a specific gene or any RNA that takes on the Z-form. Furthermore, the mechanism of Z-formation needs to be examined. Poxviruses are known to induce oxidative stress, which is known to lead to damage in nucleic acids by the introduction of the hydroxyl group into nucleotide structure (for instance, formation of 8-oxoguanine from guanine), which renders such nucleic acid more prone to flip to the Z-form [113,114,115]. Oxidative stress that is induced during poxviral infection could lead to the formation of Z-RNA, which subsequently could lead to the activation of necroptosis. Finally, it will be exciting to see how studies described in this review will contribute to the development of novel therapeutic approaches based on Z-RNA to enhance type I IFN immunity.

## Figures and Tables

**Figure 1 biomedicines-10-00235-f001:**
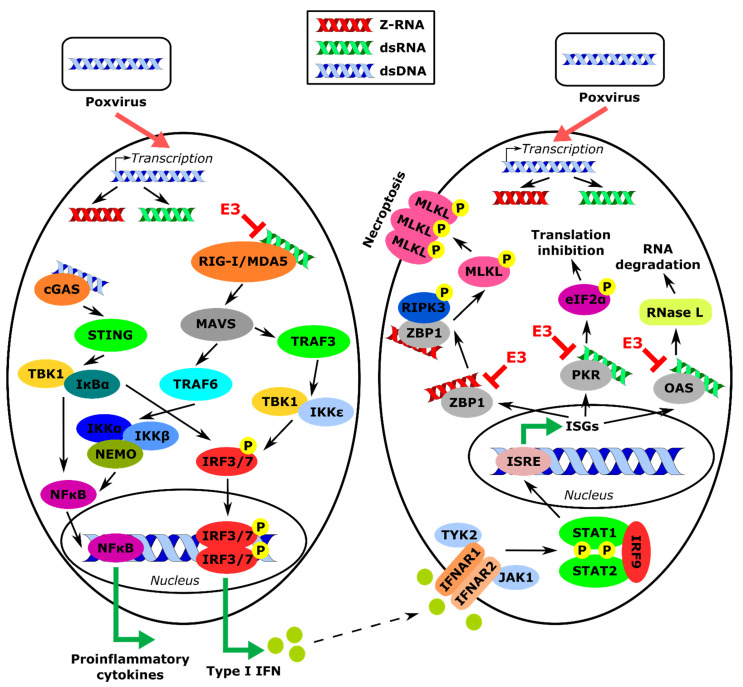
Summary of type I IFN response induction by poxviruses and its inhibition by VACV E3 protein. During poxviral infection, transcription of the viral genome results in the production of dsRNA and Z-RNA. Furthermore, dsRNA is sensed by cytosolic sensors RIG-I/MDA5 and cGAS, which induce a signaling cascade leading to the production of proinflammatory cytokines and type I IFN. IFN is sensed by neighboring cells, where it induces the expression of ISGs, including OAS, PKR, and ZBP1. Activation of OAS and PKR by dsRNA leads to RNA degradation and inhibition of translation, respectively. Activation of ZBP1 by Z-RNA results in induction of cellular death pathway necroptosis. The VACV E3 protein binds to dsRNA and thereby inhibits its recognition by RIG-I/MDA5 and cGAS (red text and red lines). E3 also inhibits activation of OAS and PKR by binding to dsRNA, and inhibits activation of ZBP1 by binding to zRNA. Abbreviations used in this figure include: eIF2α: eukaryotic translation initiation factor 2 subunit alpha; IFN: interferon; IFNAR1/2: IFN alpha and beta receptor subunit 1/2; IκBα: NFκB inhibitor alpha; IKKα/β/ε: IκBα kinase alpha/beta/epsilon; IRF3/7/9: interferon regulatory factor 3/7/9; ISG: IFN-stimulated gene; ISRE: IFN-sensitive response element; JAK1: Janus kinase 1; MAVS: mitochondrial antiviral-signaling protein; MDA5: melanoma differentiation-associated protein 5; MLKL: mixed-lineage kinase-like; NEMO: NFκB essential modulator; NFκB: nuclear factor of kappa-light-chain-enhancer of activated B-cells; OAS: oligoadenylate synthetase; PKR: protein kinase dependent on RNA; RIG-I: retinoic acid-inducible gene I; RIPK3: receptor-interacting protein kinase 3; STAT1/2: signal transducer and activator of transcription 1/2; STING: stimulator of interferon genes; TRAF3/6: tumor necrosis factor receptor-associated factor 3/6; TBK1: TRAF family member-associated NFκB activator (TANK)-binding kinase 1; TYK2: tyrosine kinase 2; ZBP1: Z-DNA-binding protein 1.

**Figure 2 biomedicines-10-00235-f002:**
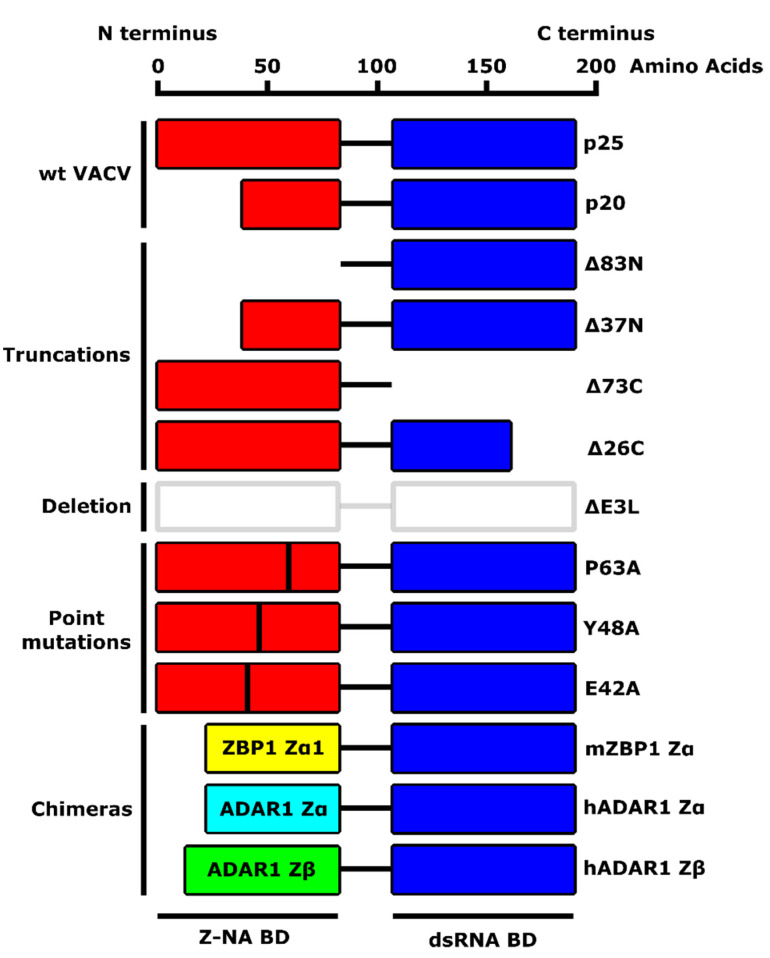
A schematic diagram of VACV E3 protein and its mutants described in this review. VACV E3 protein comprises an N-terminal Z-nucleic acid (NA)-binding domain (BD) (also referred to as Zα) and a C-terminal dsRNA BD. Wild-type VACV E3 naturally occurs in two forms: p25 (full-length protein) and p20 (N-terminal truncation, resulting from leaky-scanning translation). Described mutants include truncations (*E3L*Δ83N, *E3L*Δ37N, *E3L*Δ73C, *E3L*Δ26C), a deletion of the entire E3 protein (Δ*E3L*), single amino acid substitutions (*E3L* P63A, *E3L* Y48A, *E3L* E42A), and chimeric E3 mutants with the Zα domain substituted with homologous Z-NA BDs from Z-DNA-binding protein 1 (ZBP1) or RNA-specific adenosine deaminase 1 (ADAR1).

**Figure 3 biomedicines-10-00235-f003:**
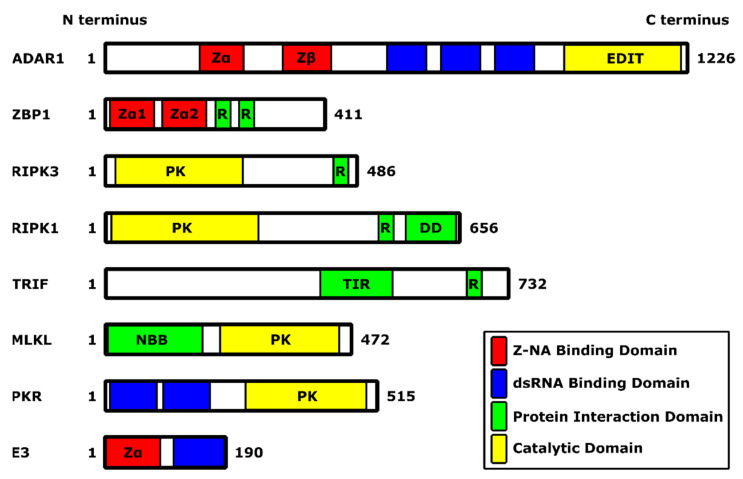
Summary schematic of functional domains present in proteins described in this review. Types of domains are indicated according to the legend. Digits indicate the first and last amino acid residue. For roles and functions of the particular protein, see the main text. Abbreviations used in this figure include: DD: death domain; EDIT: adenine-to-inosine editase; NBB: N-terminal bundle and brace; PK: protein kinase; R: receptor-interacting protein (RIP) homotypic interaction motif (RHIM); TIR: Toll/interleukin-1 receptor. Domains are depicted in scale.

**Figure 4 biomedicines-10-00235-f004:**
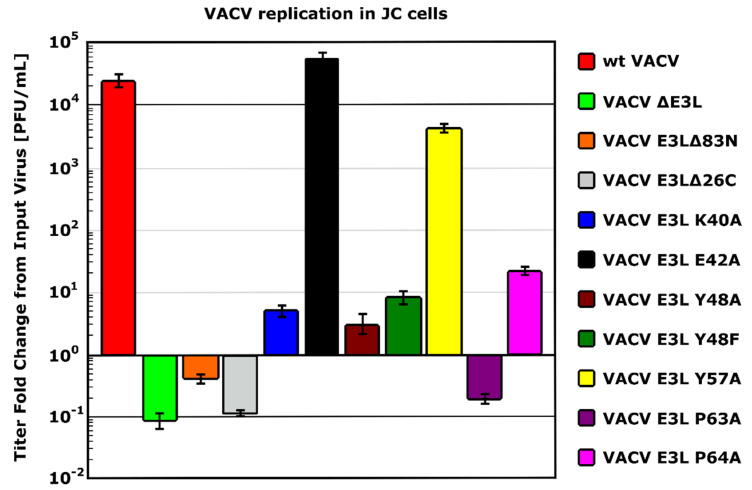
Replication of VACV expressing distinct mutants of E3 in JC cells. A multi-step growth curve was conducted for 72 h in JC cells infected at a multiplicity of infection (MOI) of 0.01. The output virus was normalized against the input virus and graphed. VACV expressing E3 mutants unable to bind to dsRNA (*E3L*Δ26C) and Z-DNA (*E3L*Δ83N, *E3L* Y48A, *E3L* P63A and partially impaired mutants *E3L* Y48F and *E3L* P64A) displayed limited ability to replicate in JC cells. Mutations in residues not associated with Z-DNA-binding (E42A and Y57A) did not appreciably affect replication in JC cells. This data demonstrates the importance of both the N- and C-terminal domains of E3 for virus replication in JC cells and it corresponds to what was seen in vivo (see the main text).

**Figure 5 biomedicines-10-00235-f005:**
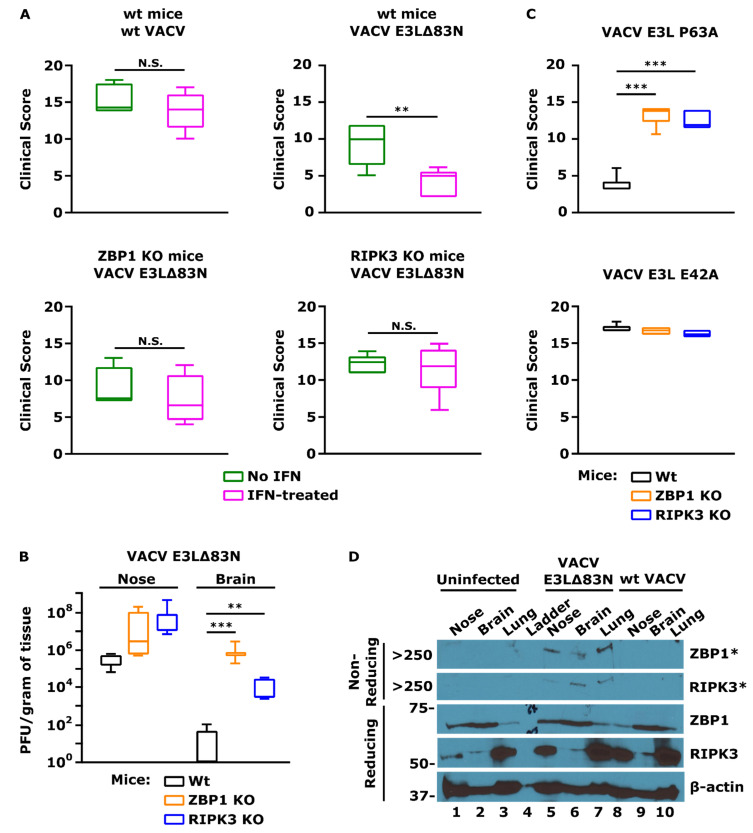
Significance of the ZBP1/RIPK3/MLKL axis for VACV pathogenesis. (**A**) Rescue of IFN resistance of VACV *E3L*Δ83N in ZBP1^−/−^ and RIPK3^−/−^ mice. C57BL/6 (wild-type or the indicated knockout) mice were infected IN with 10^6^ pfu of either wild-type VACV or VACV *E3L*Δ83N in the presence or absence of IFN. Animals were monitored for disease progression for 5 days. Clinical scores on day 5 are presented. The wt VACV is highly pathogenic and resistant to IFN treatment (top left). VACV *E3L*Δ83N displayed IFN sensitivity (top right). Knocking out the necroptosis pathway led to IFN resistance of VACV *E3L*Δ83N, as displayed by infection of ZBP1^−/−^ mice (bottom left) and RIPK3^−/−^ mice (bottom right). Student’s *t*-test *p*-values: ** *p* < 0.01; N.S. *p* > 0.05 (no statistical difference). (**B**) Rescue of VACV *E3L*Δ83N spread in ZBP1^−/−^ and RIPK3^−/−^ mice. C57BL/6 (wild-type or the indicated knockout) mice were infected IN with 10^8^ pfu of VACV *E3L*Δ83N and the indicated tissues were harvested on day 5 post-infection. The plaque assay revealed that VACV *E3L*Δ83N did not spread efficiently to the brain following IN inoculation of wt mice. Viral spread was rescued in mice with knockout of the necroptosis pathway: ZBP1^−/−^ and RIPK3^−/−^ mice. Student’s *t*-test *p*-values: ** *p* < 0.01; *** *p* < 0.001. (**C**) Rescue of VACV *E3L* P63A pathogenesis in ZBP1^−/−^ and RIPK3^−/−^ mice. C57BL/6 (wild-type or the indicated knockout) mice were infected IN with 10^6^ pfu of either Z-NA-binding mutant VACV *E3L* P63A or control point mutant, VACV *E3L* E42A. Animals were monitored for disease progression like in A. VACV *E3L* P63A displayed decreased pathogenicity in wt mice (top) as opposed to the control mutant with unaffected ability to bind to Z-NA (bottom). VACV *E3L* P63A pathogenesis was restored in ZBP1^−/−^ and RIPK3^−/−^ mice, confirming results seen for VACV *E3L*Δ83N. Student’s *t*-test *p*-value: *** *p* < 0.001 (**D**) Activation of necroptosis in mice by VACV *E3L*Δ83N. C57BL/6 mice were infected IN with 10^8^ pfu of VACV *E3L*Δ83N and the indicated tissues were harvested on day 5 post-infection. DSP crosslinking was performed as previously described [101]. Infection of mice with VACV *E3L*Δ83N resulted in the activation of ZBP1 and RIPK3 and necrosome formation (indicative of necroptosis) in the collected tissues (non-reducing gel), to which the virus was shown to spread in B. Necroptosis was not activated in mice infected with wt VACV or uninfected mice. Protein size reference is expressed in kiloDaltons (kDa). * indicates an aggregated form of the protein, present in the stacking gel, above the largest band (250 kDa) of the protein ladder.

**Table 1 biomedicines-10-00235-t001:** Summary of the phenotypes of VACV expressing given E3 protein mutants.

E3 Protein	dsRNA Binding	PKR Inhibition	Z-NA Binding	Necroptosis Inhibition	Pathogenicity	Reference
wt E3	yes	yes	yes	yes	high	[16,26,40,72]
Δ83N	yes	no	no	no	low	[16,26,40,72]
Δ37N	yes	no	no	no	low	[16,26,40,72]
Δ73C	no	no	(yes)	yes	(low)	[16]
Δ26C	no	no	(yes)	yes	low	[16,72,76]
Δ*E3L*	no	no	(no)	yes	low	[16,26,72,76]
P63A	(yes)	yes	no	no	low	[16,75]
Y48A	(yes)	yes	no	no	low	[16,75]
E42A	(yes)	yes	(yes)	yes	high	[16]
mZBP1 Zα	(yes)	(yes)	yes	yes	high	[16,75]
hADAR1 Zα	(yes)	(yes)	yes	yes	high	[16,75]
hADAR1 Zβ	(yes)	(yes)	no	no	low	[16,75]

( ) phenotype inferred from available evidence, not tested directly.

## Data Availability

The data presented in this study are available on request from the corresponding author.

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
