# Peer review of "Small Hero with Great Powers: Vaccinia Virus E3 Protein and Evasion of the Type I IFN Response"

_biomedicines, 2022, doi:10.3390/biomedicines10020235_

Round 1
Reviewer 1 Report
This manuscript by Szczerba et al. presents a comprehensive review of the various roles the protein (E3) encoded in the vaccinia virus E3L gene plays in modulating the type I interferon response to the virus’ advantage. The review is well structured and well written.
Just one minor comment: It is now more conventional to italicize the name of viral genes. In line with this norm, the authors should italicize E3L in the body of the review wherever it was used in reference to the gene encoding the vaccinia E3 protein.
Author Response
Thank you for your review and comments. Responses are in the attached file.
Reviewer 2 Report
The authors have widely covered topics on vaccinia virus E3 protein and evasion of the type I IFN response. This manuscript presents vital points regarding
- E3 protein and its mutants
- Mechanism of inhibition of IFNs activation by E3 protein
- Role of the ZBP1 in pathogenesis
Comments:
- Are Figures 4&5 and supplementary gel images are based on the previously published data? Kindly explain
- Expansion for ZBP1? z-DNA binding protein 1? (line#89)
- Line#325: viruses here, not viral system
- Line#452-454: instead of these lines, the authors can convey their own ideas and possible future out comes here.

Author Response

(The authors gave the same response as above.)

Reviewer 3 Report
This is well written review describing the various ways the vaccinia virus (VACV) E3 protein inhibits the interferon response induced upon viral infection. Most of the research in this area has been conducted by the authors’ own laboratory with various collaborators. Their research has identified two major domains of the E3 protein (N-terminal and C-terminal) that display distinct mechanisms of interference with the interferon response relying either on binding to dsRNA or binding to ZRNA, a recently discovered feature of the E3 protein. They have further demonstrated that the ZRNA binding component is responsible for VACV inhibition of necroptosis and their review highlights how this process occurs through the protein mediators of the necroptosis pathway. Finally, the authors point out a few areas where further research into our understanding of viral-induced necroptosis and the role of the E3 protein is needed. I have only one comment concerning the figures. If any figure appears in a previous publication its origin should be referenced. I gather that figure 5 presents data not previously published so this should be mentioned in the manuscript. Figure 5D is not very convincing since there appears to be no striking difference in RIPK3 and ZBP1 between the mutant virus E3L∆83N and wt VACV.
Author Response

(The authors gave the same response as above.)
